# A Self-Healing System Based on Ester Crosslinks for Carbon Black-Filled Rubber Compounds

Bashir Algaily [1,2], Wisut Kaewsakul [3,*], Siti Salina Sarkawi [4] and Ekwipoo Kalkornsurapranee [1]

1. Polymer Science and Technology, Division of Physical Science, Faculty of Science,
   Prince of Songkla University, Hat Yai Campus, Songkhla 90110, Thailand; bashir89@neelain.edu.sd (B.A.);
   ekwipoo.k@psu.ac.th (E.K.)
2. Department of Physics, Faculty of Science and Technology, Al-Neelain University, Khartoum 11111, Sudan
3. Elastomer Technology and Engineering, Department of Mechanics of Solids, Surfaces and Systems,
   Faculty of Engineering Technology, University of Twente, P.O. Box 217, 7522 NB Enschede, The Netherlands
4. Malaysian Rubber Board, RRIM Research Station, Sg. Buloh, Selangor 47000, Malaysia; ssalina@lgm.gov.my
* Correspondence: w.kaewsakul@utwente.nl; Tel.: +31-53-489-7117

**Abstract:** Carbon black-reinforced rubber compounds based on the blends of natural rubber (NR) and butadiene rubber (BR) for tire sidewall applications were formulated to investigate the self-healing efficacy of a modifier called EMZ. This modifier is based on epoxidized natural rubber (ENR) modified with hydrolyzed maleic anhydride (HMA) as the ester crosslinking agent plus zinc acetate dihydrate (ZAD) as the transesterification catalyst. The influence of EMZ modifier content in sidewall compounds on processing characteristics, reinforcement, mechanical and fatigue properties, as well as property retentions, was investigated. Increasing the content of EMZ, the dump temperatures and Mooney viscosities of the compounds slightly increase, attributed to the presence of extra polymer networks and filler–rubber interactions. The bound rubber content and Payne effect show a good correction that essentially supports that the EMZ modifier gives enhanced filler–rubber interaction and reduced filler–filler interaction, reflecting the improved homogeneity of the composites. This is the key contribution to a better flex cracking resistance and a high fatigue-to-failure resistance when utilizing the EMZ modifier. To validate the property retentions, molecular damages were introduced to vulcanizates using a tensile stress–strain cyclic test following the Mullins effect concept. The property retentions are significantly enhanced with increasing EMZ content because the EMZ self-healing modifier provides reversible or dynamic ester linkages that potentially enable a bond-interchange mechanism of the crosslinks, leading to the intermolecular reparation of the rubber network.

**Keywords:** self-reparation; crosslink; composite; sidewall compound; polymer failure

## 1. Introduction

A sidewall is one of the crucial components in a tire, which protects the tire body plies from flex fatigue under a certain dynamic mechanical load of a running vehicle plus a driver, passengers, and their belongings. Tire sidewalls are mechanodynamically stretched, bent and compressed a great number of cycles during driving. Consequently, the elastomer matrix of the sidewalls initially generates microscopic cracks/damages. Thereafter, the cracks propagate to a macroscopic level, leading to the failure of the sidewalls [1,2]. In other words, as the damages start from a molecular scale, the repeated dynamic operations of rubbers result in the progressive damages of rubber molecular networks, that grow to a larger defect scale, called macroscopic damages [3]. These initiated damages are cumulative under service circumstances and will not disappear at later stages even when an operation of rubbers, e.g., a running vehicle, stops. Therefore, the sidewalls must have good resistance to fracture growth [4]. In addition, current regulations demand tires to possess a longer lifetime and reduced fuel consumption without sacrificing car safety [5].

Thus, prevention, sometimes called extended mobility, is a solution to prolong the lifetime of tires. A self-healing ability seems to be a good route of such prevention since the cumulative damages in rubber products can be auto repaired by self-removing any local molecular damages which occur throughout their service life. An undesired accident due to a sudden explosion of a tire would be avoidable.

The self-healing aspect in polymeric materials is defined as the ability of a polymeric matrix to fully or partially recover its integrity after damages [6]. In recent decades, various intrinsic self-healing elastomers have been evolved by introducing dynamic networks to the vulcanized elastomers. These networks of vulcanizates are based on dynamic crosslinks, which are defined as, when the networks are damaged to a certain scale due to applied forces and deformations, the broken dynamic crosslinking bonds exhibit reversibility to reform the molecular networks again [7]. These dynamic bonds can be derived from, for instance, hydrogen [8] and ionic [9] bonding, Diels–Alder mechanism [10], a thiol–disulfide exchange reaction [11] and a transesterification reaction [12]. For the later, the dynamic bonds can be introduced to the elastomers using an ester crosslinking substance in combination with a catalyst to enable a transesterification reaction between ester functionality and alcohol under triggering conditions [13]. This reaction enables a thermally reversible network in a vulcanized polymer due to the exchangeable covalent bonds of ester crosslinks [14–17]. Leibler and coworkers [18] reported a successful approach of using a self-healing system capable of enabling a transesterification reaction between the epoxide groups of epoxy resins. The transesterification kinetics are three orders of magnitude; the reaction proceeds faster at elevated reaction temperatures [18]. Thus, the material can flow and is thermally reprocessable like a thermoplastic polymer while behaving as a thermoset at room temperature.

Previous work reported that epoxidized natural rubber (ENR) can be crosslinked with dicarboxylic acids [19] in the presence of zinc acetate dihydrate as a transesterification catalyst [20]. The obtainable crosslinked ENR behaves as a thermoset with an adaptable dynamic network. The catalyst initiates a transesterification reaction that introduces a dynamic network to the system, enabling the self-adhesion of ENR vulcanizates at elevated temperatures. Moreover, our previous publications [21,22] recently demonstrated that the rubber pellets crushed from a vulcanizate based on ENR crosslinked with an ester crosslinking system, i.e., a dicarboxylic acid, that is, hydrolyzed maleic anhydride, can be remolded, giving again a coherent rubber sheet. This is because the ester bonds at the interfaces of the vulcanizate pellets can exchange their bonding sites. This recreates the chemical ester crosslinks among the available active rubber chains at the pellet interfaces, leading to a reformed coherent network/matrix.

For self-healable rubbers, the key focus of most previous studies was on the restoration of localized macroscale damages, i.e., the creation of new interfaces, for instance, surface scratches, matrix cracks, which result in the subsequent loss of stiffness and strength. The optically detectable microscopic damage is considered a rather severe onset stage towards a product failure. Therefore, the healing of smaller molecular scale damages, such as chain scissions, loosened chain entanglements, broken crosslinking/interaction bonds that cause the breakage of polymer network must better prevent the material against a bulk failure [7]. In addition, the healing measurements of these molecular damages are more reproducible than the assessment of healing efficiency of completely fractured bulk materials [7]. Rubbers exhibit a considerable change in their tensile properties resulting from the first extension. This phenomenon was investigated by Mullins and coworkers [23,24]. Consequently, it led to the well-known concept of "Mullins effect or stress softening effect" [25]; this effect can typically be observed in filled [26] and unfilled rubbers [23]. The mechanism of the Mullins effect is characterized by a decrease in the stress on unloading compared to the stress on loading at the same strain. The physical phenomena taking place during the stress softening are considered damaging processes that can be repaired by a thermal or solvent-exposure treatment as triggers. Whereas, at room temperature, this healing effect is neglected [25].

This investigation aimed to explore the possibility of a modifier to make tire sidewall vulcanizates self-healable. The modifier applied was named "EMZ", which was based on ENR crosslinked with hydrolyzed maleic anhydride as an ester crosslinking agent and zinc acetate dihydrate as a transesterification catalyst. This modifier has not been reported for its application as a self-healing aid in any carbon black-filled conventional rubber composites, in this case, a sidewall compound. To validate its contribution to the self-reparation of molecular damages in sidewall compounds, various contents of the self-healing modifier were formulated into the compounds. It is worth determining if the adaptability of the ester crosslinks in this modifier enable the recombination of separate molecular networks in the vulcanizates. The restoration of network damages, such as the rupture of some physical and chemical bonds caused by multiple applied straining cycles, are verified in this investigation.

## 2. Materials and Methods

### 2.1. Materials

Natural rubber grade STR5L (Chalong Latex Industry Co., Songkla, Thailand) and polybutadiene rubber (BR01, P.I. Industries, Bangkok, Thailand) were used. Compounding ingredients included high abrasion furnace black (HAF, N330, Thai Carbon Black Public Co., Ltd., Bangkok, Thailand), treated distillate aromatic extract or TDAE oil or (H&R, Hamburg, Germany), n-1,3-dimethylbutyl)-n-phenyl-p-phenylenediamine or 6PPD (Vessel Chemical, Bangkok, Thailand), paraffin wax (Nippon Seiro Co., Ltd., Chonburi, Thailand), zinc oxide or ZnO (Thai-Lysaght Co., Ltd., Phra Nakhon Si Ayutthaya, Thailand), stearic acid (Imperial Industrial Chemicals Co., Ltd., Bangkok Thailand), n-cyclohexyl-2-benzothiazole sulfenamide or CBS, and 2,2,4-trimethyl-1,2-dihydroquinoline or TMQ (both from Lanxess Co., Ltd., Bangkok, Thailand), and sulfur (Siam Chemicals Co., Ltd., Samutprakarn, Thailand). The ingredients for the self-healing modifier based on epoxidized natural rubber (ENR) modified with a dicarboxylic acid and zinc acetate dihydrate included ENR containing 50 mol% epoxy groups or ENR-50 (Muang Mai Guthrie PCL, Phuket, Thailand), Kaolin clay (Siam Chemicals Co., Ltd., Samutprakarn, Thailand), maleic anhydride or MA (Sigma-Aldrich, Shanghai, China), 1,2-dimethylimidazole or DMI (Alfa Aesar, Kandel, Germany) and zinc acetate dihydrate or ZAD (analytical grade, Ajax Finechem, New South Wales, Australia). They were used without further purification.

### 2.2. Sample Preparation

2.2.1. Preparation of the Self-Healing Modifier

ENR-50, hydrolyzed maleic anhydride (HMA), DMI, and ZAD at 100, 3, 5, and 5 phr (parts per hundred rubber by weight), respectively, were mixed in an internal mixer for 15 min with a rotor speed of 70 rpm at a starting mixer temperature of 40 °C, following the protocol described in previous work [3,21,22]. The obtainable rubber masterbatch was named "EMZ" (epoxidized natural rubber modified with hydrolyzed maleic anhydride plus zinc acetate dihydrate) as an acronym in this context. Regarding the function of the ingredients used, DMI was employed as an accelerator for the esterification reaction between the carboxylic acids of HMA and epoxide groups of ENR. It was manually premixed with kaolin clay, an inert filler, prior to introduction into the mixer to facilitate the incorporation of the DMI into the compound. ZAD functions as a transesterification catalyst with efficiency to promote the transesterification reaction at elevated temperatures, shuffling the formed ester bonds and enabling network rearrangement [20]. This mechanism allows the crumbled rubber to again reform into a coherent vulcanized sheet [21,22]. Hence, this material is considered a self-healing modifier, as it would potentially promote the intermolecular self-reparation of molecular damages in vulcanizates. The material structure, cure characteristics, and other technical properties of this modifier were recently reported in [21,22].

### 2.2.2. Compound Preparation

The formulation of the sidewall compounds used for this study is shown in Table 1. Two sets of the compounds were prepared with two different EMZ additions—Set A: extra addition of EMZ at 0, 5, 10 and 15 phr to the compounds coded as S0, E-S5, E-S10, and E-S15, respectively; and Set B: blending of rubber/EMZ (i.e., the total amount of rubber is 100 phr) at the ratios of 100/0, 95/5, 90/10 and 85/15, giving the compounds, namely S0, B-S5, B-S10, and B-S15, respectively. The reason for designing these two different series for the present investigation is that the EMZ modifier is based on ENR. Thus, it can be considered as either a compatibilizer or a secondary elastomer. A compatibilizer is commonly added as an extra ingredient to compounds, while rubber is added by blending with the base rubber of compounds.

**Table 1.** Compound formulations used in this study.

| Ingredients | Dosage (Phr) | | | | | | |
|---|---|---|---|---|---|---|---|
| | Ref. | Set A: Extra Addition of EMZ [a] | | | Set B: Blending of EMZ [b] | | |
| | S0 | E-S5 | E-S10 | E-S15 | B-S5 | B-S10 | B-S15 |
| NR [c] (STR 5L [d]) | 50 | 50 | 50 | 50 | 47.5 | 45 | 42.5 |
| BR [e] | 50 | 50 | 50 | 50 | 47.5 | 45 | 42.5 |
| EMZ [f] | 0 | 5 | 10 | 15 | 6.2 | 12.3 | 18.5 |
| Carbon black (HAF, N330 [g]) | 50 | | | | | | |
| Process oil (TDAE [h]) | 10 | | | | | | |
| TMQ [i] | 1 | | | | | | |
| 6PPD [j] | 2.5 | The quantities of these compositions are the same with those added for the reference (Ref.) compound (S0). | | | | | |
| Paraffin Wax | 1.5 | | | | | | |
| Zinc oxide | 4 | | | | | | |
| Stearic acid | 2 | | | | | | |
| CBS [k] | 2 | | | | | | |
| Sulfur | 2.5 | | | | | | |

[a] The total rubber contents of these compounds were higher than 100 phr due to the particular extra amount of EMZ masterbatch added to the compounds.; [b] The total rubber contents of these compounds were at 100 phr.; [c] natural rubber; [d] standard Thai rubber grade 5L; [e] polybutadiene rubber; [f] epoxidized natural rubber modified with hydrolyzed maleic anhydride plus zinc acetate dihydrate; [g] high abrasion furnace grade N330; [h] treated distillate aromatic extract; [i] 2,2,4-trimethyl-1,2-dihydroquinoline; [j] n-(1,3-dimethylbutyl)-n-phenyl-p-phenylenediamine; [k] n-cyclohexyl-2-benzothiazole sulfenamide.

Mixing was performed using an internal mixer (Charoen Tut Co., Ltd., Samutprakarn, Thailand). The internal mixer was operated at a rotor speed of 100 rpm and a fill factor of 70%. The initial temperature of the mixer was adjusted to 50 °C. The dump temperature of all compounds was recorded in the range of 110–130 °C. Firstly, NR was masticated for 1 min to reduce its viscosity, then BR and EMZ were added and mixed for another 1 min prior to the addition of carbon black and process oil. The mixing was continued for another 3–4 min until a stable ram torque was obtained. Other chemical ingredients, i.e., ZnO, stearic acid, TMQ, 6PPD and paraffin wax were subsequently added and mixed for 2 min. Immediately after dumping, the obtainable compound was then sheeted out on a two-roll mill and kept overnight before adding CBS and sulfur on a two-roll mill. The final compounds were sheeted and kept overnight prior to their property characterizations.

### 2.3. Characterizations

#### 2.3.1. Cure Characteristics and Mooney Viscosity

The cure characteristics of the compounds were determined using a moving die rheometer or MDR (MDR2000, Alpha Technologies, OH, USA) with a curing temperature of 150 °C for 30 min. The optimum cure time ($T_{c90}$) and rheometer cure torque were reported and used for press-vulcanizing the compounds into 1.5 mm-thick sheets. The investigated compounds were tested for their Mooney viscosity using a Mooney viscometer

(MV2000, Alpha Technologies, Alpha Technologies, OH, USA) at 100 °C with a large rotor according to American Society for Testing and Materials or ASTM D1646. The value of ML (1 + 4) 100 °C was reported. Note that ML stands for Mooney (M) viscosity tested using a large (L) rotor.

### 2.3.2. Apparent Crosslink Density

The measurements of the apparent crosslink density of vulcanizates were carried out via a swelling method with the vulcanizates in toluene as solvent according to ASTM D471–12a. The cured test specimens of $15 \times 15 \times 1.5$ mm$^3$ were immersed in 30 mL of toluene for 7 days at room temperature. The samples were removed from the toluene, and then the weight of the swollen samples was measured. The specimens were dried in a vacuum oven at 75 °C for 24 h, and the dry weight of the specimens was finally measured. The apparent crosslink density ($\nu$) was calculated using the modified Flory–Rehner equation as shown in Equation (1) [27]:

$$\nu = \frac{-(\ln(1 - V_{r0}) + V_{r0} + \chi V_{r0}^2)}{2V_s \left( V_{r0}^{\frac{1}{3}} - \frac{V_{r0}}{2} \right)} \tag{1}$$

where $V_s$ is the molar volume of the toluene (i.e., 106.9 cm$^3$/mole), $\chi$ is the Huggins interaction constant the interaction parameter between the BR and the toluene is 0.340, and the interaction coefficient between the NR and the toluene is 0.393 [28]. However, for the materials used in this study, NR/BR are 50/50 blended compounds, so it is possible to calculate the interaction coefficient to be 0.367 [28]. However, $V_{r0}$ is the volume fraction of the swollen rubber, which could be computed using Equations (2) and (3) [28]:

$$\frac{V_{r0}}{V_{rf}} = 1 - m\frac{\theta}{1 - \theta}, \tag{2}$$

$$m = 3c\left(1 - V_{r0}^{\frac{1}{3}}\right) + V_{r0} - 1 \tag{3}$$

where $c$ is the parameter for a given filler; $c$ = 1.17 for carbon black, $\theta$ is the volume fraction of the filler, and $V_{rf}$ is the volume fraction of rubber in the swollen filled rubber, which can be calculated using Equation (4):

$$V_{rf} = \frac{\frac{w_2}{\rho_r}}{\frac{w_2}{\rho_r} + \frac{(w_1 - w_2)}{\rho_s}} \tag{4}$$

where $w_1$ is the weight of a swollen sample, $w_2$ is the weight of the sample after drying, and $\rho_s$ and $\rho_r$ are the densities of the solvent used (i.e., 0.886 g/cm$^3$ for toluene) and of the rubber (i.e., 0.92 g/cm$^3$ for natural rubber), respectively.

### 2.3.3. Bound Rubber Content as Indicative of Filler–Rubber Interaction

Bound rubber content was measured using uncured samples without curatives. The measurement was carried out under an ammonia atmosphere in order to cleave physical linkages in the samples, giving only the amount of chemically bound rubber content, i.e., filler–rubber interactions, as well as the lightly crosslinked rubber network [21]. The procedure started with preparing an uncured sample of about 0.2 g. Then, the sample was cut into small pieces and placed in a filter bag made from a metal mesh, i.e., 400 mesh. A bag containing the sample was immersed into 20 mL of toluene at room temperature for 72 h, and the solvent was renewed every day. Afterward, the sample was removed from the toluene and dried for 24 h at 105 °C. Then, the sample was again immersed in 20 mL of toluene at room temperature for 72 h and placed in an ammonia atmosphere; the solvent was renewed every day. Finally, the sample was dried for 24 h in a vacuum hot-air oven at 105 °C. The bound rubber content of each sample was calculated according to Equation (5) [29]:

$$Bound\ rubber\ content\ (\%) = \frac{m_2 - m_1 w_2}{m_1 w_1} \tag{5}$$

where $m_1$ is the initial weight of a sample, $m_2$ is the final weight of the dried sample, $w_1$ is the rubber fraction in the compound, and $w_2$ is the filler fraction in the compound.

### 2.3.4. Tensile Properties

Vulcanizates with a thickness of about 1.5 mm were die-cut into dumbbell-shaped specimens. The tensile tests were carried out using a universal testing machine (Instron 3365, Instron Co. Ltd., Bangkok, Thailand) with a crosshead speed of 500 mm/min according to ASTM D 412 at room temperature. Five specimens per vulcanizate were performed, and the average value was reported.

### 2.3.5. Payne Effect as Indicative of Filler–Filler Interaction

The Payne effect was measured using a rubber process analyzer (RPA, Alpha Technologies, OH, USA). An uncured sample was first vulcanized directly in the RPA at 150 °C for its respective $T_{c95}$ and subsequently cooled to 100 °C prior to the Payne effect test. The test was set with a strain sweep in the range of 0.56–100% at a frequency of 0.50 Hz. The Payne effect was calculated using the data of the storage moduli measured, i.e., the difference between the storage shear moduli at 0.56 ($G'_{0.56}$) and 100% strain ($G'_{100}$).

### 2.3.6. Flex Cracking Resistance

The flex cracking resistance measurement of vulcanized compounds was performed with a De Mattia flex cracking tester (TOYOSEIKI 255, Toyo Seiki Seisaku-sho, Ltd., Tokyo, Japan) according to ISO 132 with a testing condition of 66% pre-straining of a sample, 300 bending cycles/min and room temperature (23 ± 1 °C). For each vulcanizate, six specimens were tested, and the average flex-resistance value was reported. This value was determined by the number of flexing cycles required to reach grade 3 and the mean flex cracking resistance was recorded.

### 2.3.7. Fatigue Resistance

A De Mattia flexing tester as employed for determining the flex fatigue resistance of the investigated vulcanizates was used with an extension mode. Dumbbell test specimens were prepared from a die-cut type C according to the test method ASTM D412. The specimens were subjected to a tensile straining deformation of approximately 130% relative to the non-deformed stage. In this experiment, at least four specimens of each compound were tested and the average value of the number of cycles that causes the material failure determined by the complete rupture of a test specimen was recorded.

The fatigue characteristic is relatively crucial for the tire sidewalls. Thus, apart from the Mullins effect analysis, as described in the following section, the healing efficiency of the samples was also assessed using this fatigue characterization. A tension fatigue test was conducted using a dumbbell specimen. The molecular damages were introduced to pristine samples by applying 10 multiple stress–strain cycles. After that, the damaged samples were rested to relax at room temperature for 30 min and then thermally annealed at different temperatures (i.e., room temperature, 80 and 120 °C) for another 30 min before resting to cool down to room temperature for 60 min. Finally, the fatigue tests were again conducted for the treated samples. The average number of cycles that cause the complete failure separation to the samples was reported. The values of the self-healed samples were compared with their pristine counterparts to appraise the self-healing performance.

### 2.3.8. Assessing the Self-Reparation of Molecular Damages in Vulcanizates

In general, the "damage" of a material is defined as an undesirable failure occurring in a system that affects its performance; hence, an understanding of the damage nature and the damage progression is essential to suppress the damages, maintaining high retention of the product properties. The damages do not necessarily lead to a sudden entire loss of material

performance [30], but rather a continual deterioration in overall properties, once operated under actual or simulated service conditions. The degree of damage can be evaluated by comparing the two states of a material, i.e., pristine or undamaged versus damaged.

The Mullins effect measurement is considered as an approach for monitoring the damage and recovery degree since it induces the rupture of physical and chemical bonds [25] inside a material matrix during loading cycles; the rupture immediately progresses after the first loading cycle [31]. This rupture mechanism occurs in all cases, i.e., unfilled and filled or reinforced rubbers; this phenomenon is more pronounced in reinforced rubbers [25]. However, in this study, the damage was observed as a reduction in the material stiffness as a result of microscale defects, e.g., polymer chain and network breakages [32].

By utilizing the Mullins effect concept for the present experiment, the molecular network damages were introduced by applying multiple stress/strain cycles (i.e., 10 cycles) to the samples [33]. The vulcanizates with a thickness of about 1.5 mm were die-cut into dumbbell-shaped specimens. A tensile testing machine was used; the initial free length between the clamps was 40 mm. A sample was stretched to its maximum extension ratio $\lambda$ of 70% relative to the elongation at break of the vulcanizate predetermined with a different sample at a constant crosshead speed of 100 mm/min in ambient conditions. After 10 stretching cycles, the sample was rested at room temperature without pressure for 30 min. Then, the sample was treated for 30 min at different temperatures, i.e., room temperature, as well as 40, 80, and 120 °C, in a vacuum hot-air oven and cooled down to room temperature for 60 min. The stress–strain cyclic test was repeated for the treated samples. The obtained results were compared with that of the pristine counterparts to evaluate the self-reparation degree.

In addition, the hysteresis calculated from the area of the first stress–strain cycle was assessed to indicate the change in energy dissipation of vulcanizates [34]. To quantify the recovery degree, the recovery ratio (R) was defined as the value of the hysteresis of a healed sample at a specific temperature ($A_{healed}$) to that of the original ($A_{pristine}$) one, expressed as follows:

$$R = \frac{A_{healed}}{A_{pristine}} \times 100. \tag{6}$$

## 3. Results and Discussion

### 3.1. Compound Viscosities, Cure Characteristics and Intermolecular Networks

Figure 1 shows the increase in compound viscosities determined by the Mooney viscosity and the minimum cure torque ($S'_{min}$) upon increasing the EMZ content for both series of the compounds. This was a first hint implying that the EMZ modifier might generate some intermolecular interactions inside the mixes. The mixing temperatures also rise with the increment of EMZ content. It is in line with the increase in compound viscosities because it reflects that the shear rate during mixing increases, attributed to three-dimensional networks or interactions occurring in the system. The viscosities of the compounds prepared with blending the EMZ modifier are slightly higher than the counterparts with an extra addition of the EMZ. This is due to a higher amount of the modifier in the series 'blending'. Regarding the types of possible interactions in the system, it will be discussed later in subsequent sections.

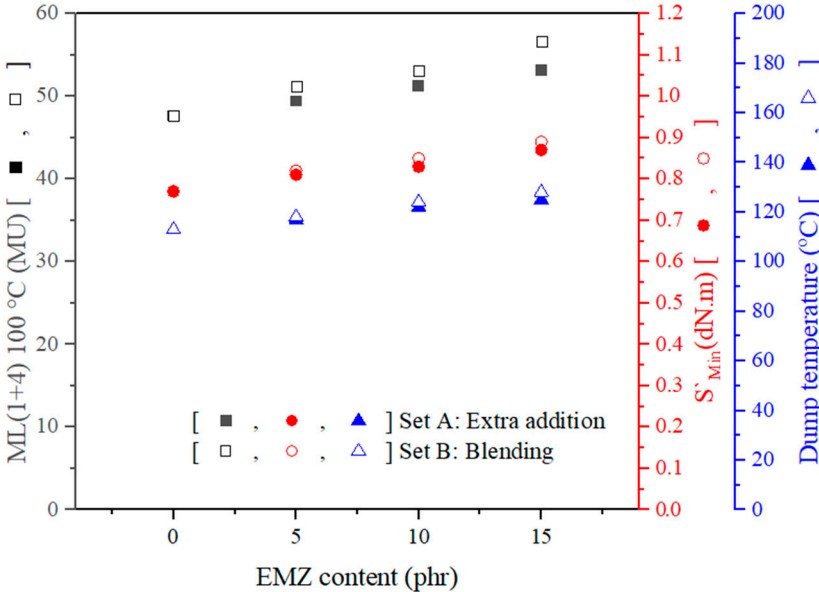

**Figure 1.** Mooney viscosity, minimum cure torque and dump temperature of carbon black-filled NR/BR compounds as a function of EMZ content added via extra addition (Set A) or blending (Set B).

The cure rate of compounds increases upon increasing the EMZ content, as displayed in Table 2. This is attributed to the additional curing substance in the system derived from the EMZ modifier. In addition, the epoxide groups of ENR, the base rubber for the modifier, can boost a sulfur-vulcanization reaction [35]. The rheometer cure torque differences (see Table 1) are in good agreement with the above elucidation regarding more intermolecular interactions taking place in the compounds. However, the cure torque also involves the effect of filler–filler interaction as discussed later in Figure 2. The Payne effect result, an indicative for the extent of filler–filler interaction, shows a significant decrease in values with increasing the EMZ content, implying that the filler–filler interaction or filler network is reduced. The lower filler–filler interaction should lessen the cure torque due to the hydrodynamic effect. However, the torque difference still shows a slight increase in values. Therefore, these results support well the elucidation of extra crosslinks generated by the EMZ modifier in a sulfur-crosslinked network.

**Table 2.** The cure characteristics of carbon black-filled NR/BR compounds with extra addition (E-Sx) and blending (B-Sx) of the EMZ modifier at different quantities (x phr).

| Compounds | Cure Characteristics | | | | | |
|---|---|---|---|---|---|---|
| | $T_{S2}$ [a] (Min) | $T_{C90}$ [b] (Min) | $S'_{Min}$ [c] (dN.m) | $S'_{Max}$ [d] (dN.m) | $S'_{Max}$–$S'_{Min}$ [e] (dN.m) | CRI [f] |
| **S0** | 0.58 | 4.39 | 0.77 | 13.23 | 12.46 | 26.2 |
| **E-S5** | 0.58 | 4.32 | 0.81 | 13.30 | 12.49 | 26.7 |
| **E-S10** | 0.49 | 3.52 | 0.83 | 13.55 | 12.72 | 33.0 |
| **E-S15** | 0.40 | 3.47 | 0.87 | 13.96 | 13.09 | 32.6 |
| **B-S5** | 0.56 | 3.95 | 0.82 | 13.98 | 13.16 | 29.4 |
| **B-S10** | 0.38 | 3.49 | 0.85 | 14.28 | 13.43 | 32.2 |
| **B-S15** | 0.36 | 3.35 | 0.89 | 15.21 | 14.32 | 33.4 |

[a] Scorch time; [b] optimum cure time; [c] minimum cure torque; [d] maximum cure torque; [e] cure torque difference; [f] cure rate index.

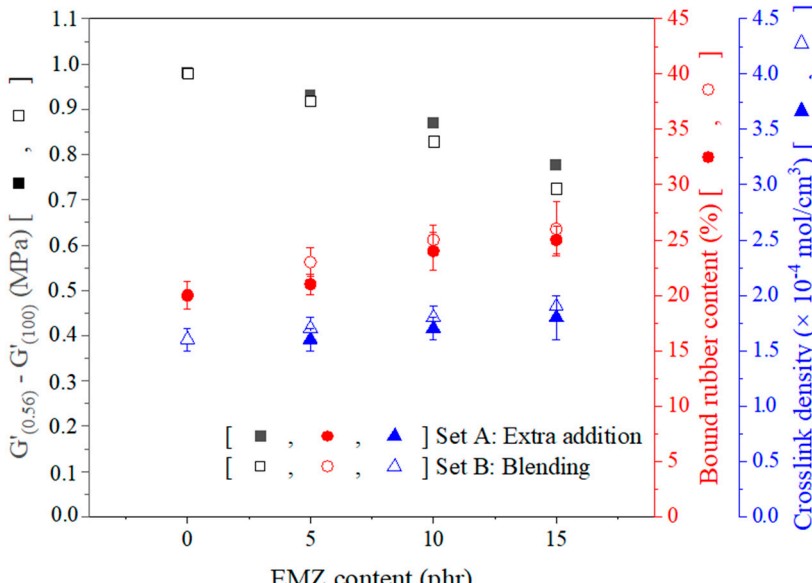

**Figure 2.** Influence of EMZ loading on the Payne effect, chemically bound rubber content, and the crosslink density of carbon black-filled NR/BR compounds. EMZ was added via two approaches: extra addition (Set A) and blending (Set B).

To verify these extra intermolecular networks, the quantities of bound rubber and apparent crosslink density were determined. Figure 2 presents the results of these two indicators. The bound rubber content increases upon increasing the EMZ loading. This filler–rubber interaction can be generated during mixing due to the penetration of rubber chains into carbon black aggregates/agglomerates, giving physically interlocking interactions between the carbon black surface and the rubber chains. Thus, the bound rubber can be detected in this stage. Increasing the EMZ content leads to a reduced Mooney viscosity of the compounds, which is consistent with a finding reported in previous work [3]. This is attributed to EMZ which contains a strong acidic substance that has the capability of peptizing rubber molecules resulting in shortened molecular chains or reduced molecular weight of the rubber matrix. The lower compound viscosity has a beneficial effect on filler dispersion since the rubber molecules can penetrate better into the aggregates as a consequence on the breakage of filler clusters leading to a better micro-dispersion of the fillers. This mechanism has been elaborated in the literature [36].

With the lower Payne effect that can be used as an indirect indicator for a better micro-dispersion, the surface area of carbon black dispersed in the rubber matrix is increased, leading to a greater potential of rubber molecules to interact with the carbon black particles. As a result, the bound rubber content rises. Apart from that, the EMZ modifier itself has dicarboxylic acid that can form the crosslinks by reacting with the hydroxyl groups from the opened oxirane structure of ENR, giving ester crosslinks at a temperature as low as 120 °C. Hence, the mixing temperature (see the dump temperatures shown in Figure 1) at about 120–130 °C can cause the ester crosslinks to take place. Owing to the extra intermolecular interactions due to the rubber-carbon black reinforcement mechanism and ester crosslinks in the EMZ phase, thus, the apparent crosslink densities of the vulcanizates slightly increase upon increasing the EMZ loading.

### 3.2. Mechanical and Fatigue Properties

Figure 3 displays the tensile strength and elongation at break of the vulcanizates with varied amounts of EMZ modifier. The tensile strength increases with increasing the EMZ content. Surprisingly, the elongation at break also has an increasing trend. However, when considering the obtainable vulcanizates, the ones with a higher EMZ content exhibit a substantially reduced Payne effect. The improvement in the Payne effect or lower filler–

filler interaction leads to a better micro-dispersion of carbon black and can also reduce micro-defects in the vulcanizates caused by larger filler agglomerates. Thus, fewer weak points in the materials result in superior elongateability. The increase in the tensile strength could be due to the higher elongation at break, yielding materials that can withstand higher loads at the ultimate stage. Moreover, the filler–rubber interaction plus a higher crosslink density when increasing the EMZ content contributes to the superior homogeneity of the filled compounds. The strong links between filler and rubber promote a better force distribution within an elastomeric composite system, as a consequence in a lower energy dissipation—the higher elasticity. Therefore, the tensile strength of the vulcanizates with higher EMZ contents is enhanced.

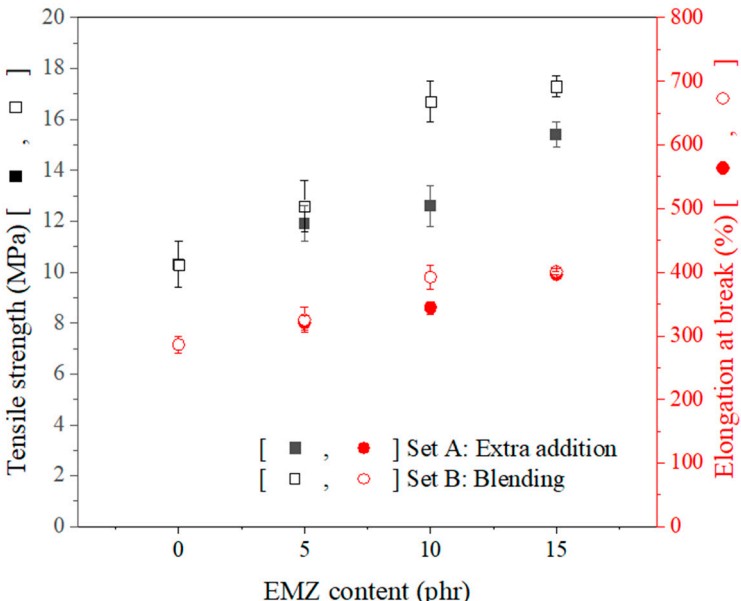

**Figure 3.** Effect of EMZ loading on the tensile strength and elongation at break of carbon black-filled NR/BR vulcanizates. EMZ was added by two approaches: extra addition (Set A) and blending (Set B).

Figure 4 illustrates the decrease in reinforcement index (M300/M100) of vulcanizates with the increased amounts of EMZ modifier. While the tensile moduli at 100 and 300% elongation slightly increase. This is because, by introducing more amounts of the EMZ modifier, the micro-dispersion as indicated by the Payne effect is improved, resulting in a more compact filler of aggregates/agglomerates in the rubber matrix, and therefore, the risen moduli or hardness of the vulcanizates. However, the better carbon black dispersion is derived from an increment in the EMZ modifier. As aforementioned, this modifier has the capability of reducing the viscosities of the compounds because it causes the chain scission of rubbers—a reduced molecular weight. This phenomenon seems to have a negative effect on the reinforcement index (M300/M100), even though the micro-dispersion is ameliorated. Therefore, it would be worth remarking that the EMZ modifier should be used at an optimum amount, which can balance the overall properties of the compounds. This appropriate amount must be tailored and applied depending on a particular rubber product.

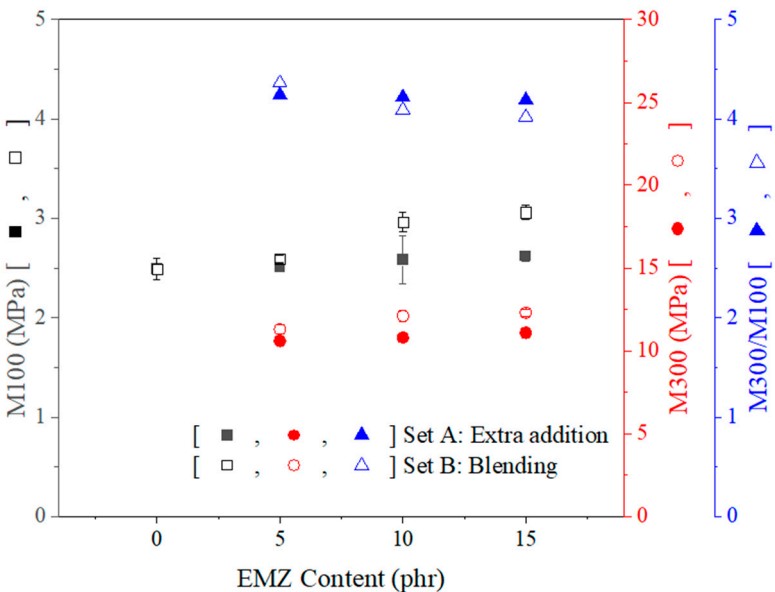

**Figure 4.** Effect of EMZ loading on the tensile moduli at 100 and 300% and the reinforcement index (M300/M100) of carbon black-filled NR/BR vulcanizates. EMZ was added by two approaches: extra addition (Set A) and blending (Set B).

The mean flex cracking resistance was determined by the number of cycles to reach the crack level of Grade 3 of the test. The crack fatigue result can qualitatively determine the service life of rubber composites [37]. Figure 5 shows the flex cracking resistance of the investigated composites in this study. It can be seen that the flex resistance of the samples is improved by the addition of the EMZ modifier. In a reinforced elastomer composite, active fillers and their interactions with rubber are the key parameters determining this property. In this system, all compounds are included with the same carbon black grade, i.e., N330, which is a conventional reinforcing filler. However, based on the Payne effect and bound rubber content (Figure 2), adding the EMZ modifier significantly reduces filler–filler interaction and increases filler–rubber interaction. This crack resistance result is in line with these two indicators as well as the tensile strength (Figure 3). Even though an NR/BR blend has a high affinity towards carbon black due to their polarity matching [38], however, the carbon black particles are not stable during compound storage and tend to form large agglomerates, because of the self-association of their active surface [39]. The large filler clusters forming the filler network cause a poor dispersion of the filler in rubber compounds, resulting in reduced flex cracking resistance [40]. Based on this fact, the primary parameter affecting the crack resistance is the homogeneity of a compound system [41]. The high crack resistance of the compound with 15 phr EMZ blending reflects a better homogeneous distribution of carbon black in rubber blends compared to the compounds with a lower content of EMZ. However, the crack propagation occurs more rapidly in vulcanized rubbers with lower flexibility. The damages of vulcanizate occur on a small molecular scale. Basically, the crosslinks from short or stiff linkages, e.g., carbon–carbon, di-sulfidic, mono-sulfidic and ester crosslinks would fail faster than, e.g., polysulfidic crosslinks. Interestingly, the elongation at break of the vulcanizates with a higher EMZ content might also relate to a better crack growth resistance. When considered at a small molecular scale, the rubber molecular network needs to withstand a very high deformation. The higher elongation reflects that the vulcanizate can withstand a higher deformation also at a small molecular scale. Therefore, this could be another reason elucidating why the crack propagation is enhanced and aligns with the elongation at break result, in addition to the better force distribution of the composites due to a higher extent of filler–rubber interaction.

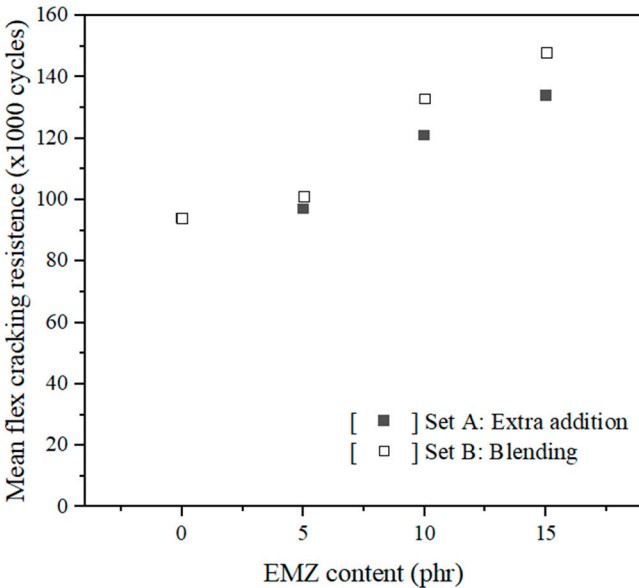

**Figure 5.** Flex-cracking resistance of carbon black-filled NR/BR vulcanizates as a function of EMZ loading. EMZ was added via two approaches: extra addition (Set A) and blending (Set B).

Furthermore, the composites with the higher contents of the EMZ modifier show a better crack resistance. This improvement in the crack propagation resistance can be attributed to the increment of sacrificial bonds (i.e., hydrogen and ionic bonds) due to the addition of the EMZ modifier, as verified in previous work [3]. The sacrificial bonds are defined as the weak crosslinks that rapture before the failure of strong covalent bonds during deformation [42]. The rapture of sacrificial bonds dissipates a huge amount of energy, which helps maintain the overall integrity of the materials [43]. The breakage of the weak network of the EMZ formed by the reversible hydrogen or ionic linkages provides an extra energy dissipation for the vulcanizates during a dynamic deformation, i.e., testing. Since this network is reversible and can re-associate after the breakage, therefore, a huge energy dissipation created by the EMZ inhibits the crack initiation and slows down the crack growth leading to the remarkable crack resistance of the vulcanizates.

*3.3. Self-Healing Performance*

The property retention of the vulcanizates was assessed via monitoring the Mullins effect. Figure 6 shows that increasing the amount of EMZ modifier significantly improves the hysteresis retention of the vulcanizates. In addition, the treatment temperature has been shown to be a relevant parameter enhancing the property retentions. This result is good evidence to support the efficiency of the EMZ modifier in enabling the self-healing of the carbon black-filled NR/BR compounds. The modifier possesses an ester crosslinking system which is based on the reaction between ENR and dicarboxylic acid. This ENR-based modifier can co-cure the rubber matrix since the ENR-50 still consists of allylic positions that are reactive towards sulfur vulcanization reaction. Therefore, the vulcanized blend of NR/BR and ENR-modifier is considered a homogenous matrix or a non-separate phase system.

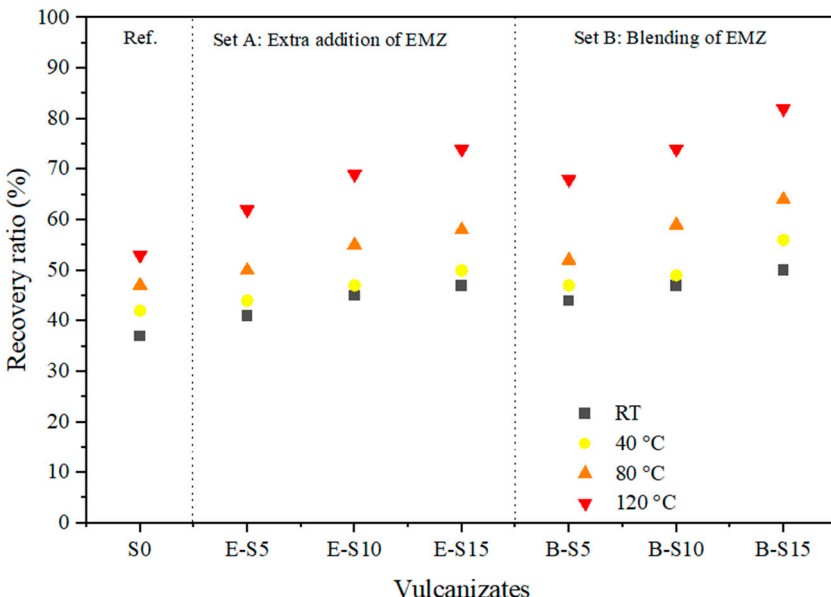

**Figure 6.** Hysteresis retention after a tensile cyclic test (i.e., 10 cycles) of carbon black-filled NR/BR vulcanizates with different contents of EMZ modifier added via either extra addition (Set A) or blending (Set B).

Inside the matrix, the modifier promotes more reversible interactions. These interactions include polar–polar linkages, hydrogen bonding and ionic crosslinks. Importantly, a transesterification reaction of the ester crosslinks can also occur through the aid of ZAD and a thermally annealing treatment [17,20,44]. Figure 7 illustrates a graphical mechanism of the exchangeable bonds of ester crosslinks to auto-repair the broken network in the vulcanizate matrix. Once the matrix undergoes molecular scale damages at, i.e., micro- and macro-levels, the interfaces of the damages could have been healed via the transesterification reaction yielding exchangeable ester crosslinks at the interfaces. The new ester crosslinks from different positions could shuffle to chemically bridge at the broken interfaces, leading to the intermolecular self-reparation of the damages. This crosslink-shuffling mechanism is accelerated upon increasing the annealing treatment temperature, as the samples show the highest property retention at the maximum temperature used, i.e., 120 °C. These results are well in line with the findings from previous studies [3,18,21,22].

Figure 8 presents the fatigue resistance of the vulcanizates. This property is considered highly important for sidewall compounds since they generally undergo hash dynamic mechanical conditions when tires roll on the road to support a vehicle. The fatigue values demonstrate the same fashion as the result of hysteresis retention (see Figure 6). It is interesting to visualize that the vulcanizate based on the blend of 15 phr of EMZ is nearly comparable to the pristine sample when annealed at 120 °C.

Therefore, it is clear that tire sidewalls containing carbon black as reinforcing filler can be precluded from molecular damage propagation by using the EMZ modifier. This feature is beneficial to improve the durability of tire sidewalls, which ameliorate the service life of tires and potentially prevent an accident caused by the sudden failure of tires during running on the road.

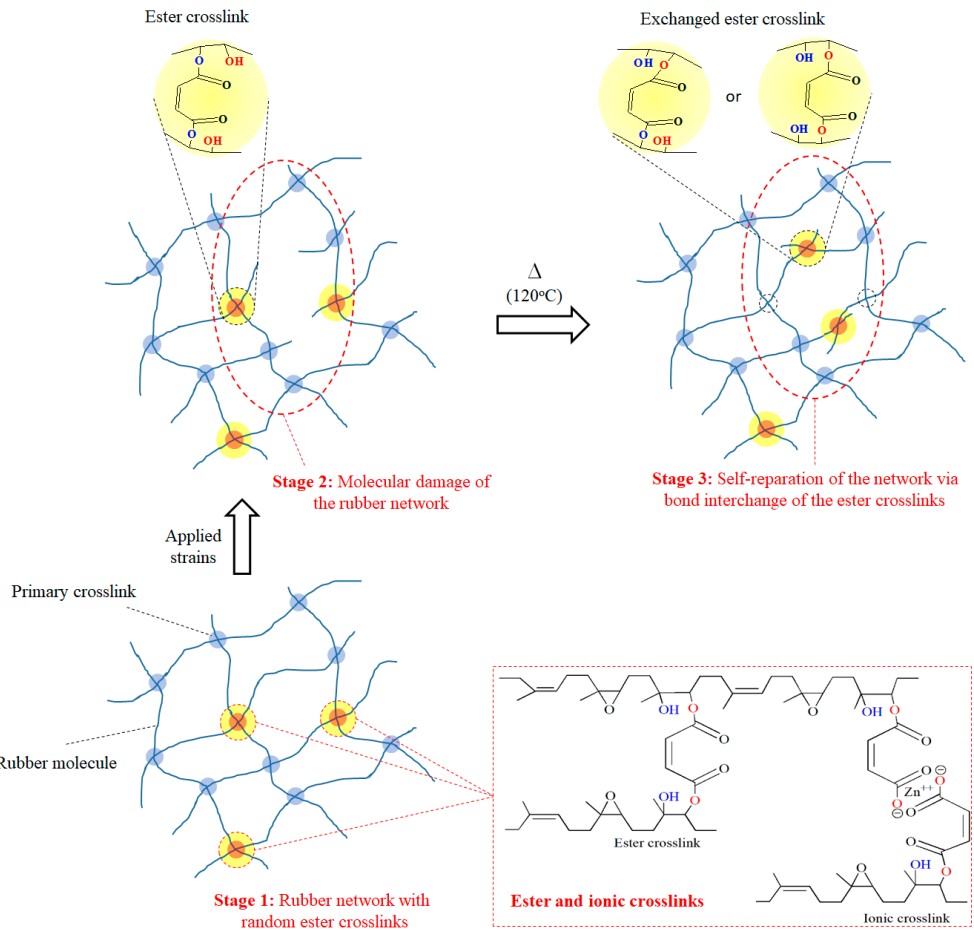

**Figure 7.** Plausible self-healing mechanism of rubber network through the transesterification reaction of ester crosslinks.

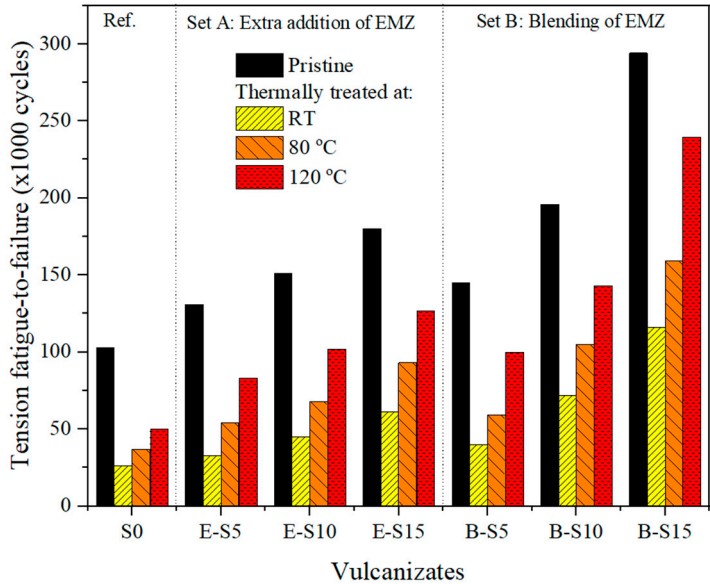

**Figure 8.** Fatigue-to-failure property of carbon black-filled NR/BR vulcanizates with different contents of EMZ modifier added via extra addition (Set A) or blending (Set B). For the pristine sample, the test was conducted before creating molecular damages, while the molecular damaged samples were tested after applying multiple tensile cycles and then thermal treatment at different temperatures.

## 4. Conclusions

The rubber composites investigated in the present study were based on blends of NR and BR, while the EMZ self-healing modifier at different amounts was added to the compounds via extra addition or blending. Increasing the content of EMZ, the dump temperatures and Mooney viscosities of the compounds slightly increased. This is because the extra polymer networks and filler–rubber interaction were generated due to the presence of the modifier, which was confirmed by the apparent crosslink densities and bound rubber contents, respectively. In addition, the EMZ could improve the micro-dispersion of carbon black indicated by the lower Payne effect values upon increasing the EMZ content. The addition of EMZ by blending gives a slightly higher impact on the overall properties than that of the extra addition to the compounds, attributed to a higher concentration of EMZ relative to the total rubber content for the blending approach. The strength properties and the moduli at both 100 and 300% strain of vulcanizates are enhanced with increasing EMZ content. The property retentions of vulcanizates are significantly improved upon increasing the EMZ content and the annealing treatment temperature. This is attributed to the EMZ contributing to more concentrations of the thermochemically reversible ester crosslinks in the system. The ester bonds can potentially be interchanged through a transesterification reaction mechanism, which enables the rearrangement of chemical ester crosslinks in the rubber network and leads to intermolecular self-reparation of the damaged network. The intermolecular self-healing mechanism of this modifier requires a thermal treatment and a transesterification catalyst after unloading conditions. The significant enhancement in flex fatigue resistance when using the EMZ modifier has a positive impact on the durability of tire sidewalls. Further investigations into the self-reparation of molecular network damages could be highly interesting and are recommended. With the unique self-repairing ability of the present system, a future generation of tire sidewalls with ameliorated durability and reduced road accidents due to tire explosions could be anticipated.

**Author Contributions:** Conceptualization, W.K. and B.A.; methodology, B.A. and W.K.; microstructural analysis, B.A.; investigation, B.A. and W.K.; funding acquisition, E.K. and W.K.; project administration, W.K. and E.K.; supervision, W.K. and E.K.; writing—original draft preparation, B.A.; writing—review and editing, W.K., E.K. and S.S.S. All authors have read and agreed to the published version of the manuscript.

**Funding:** This research was funded by the Higher Education Research Promotion and Thailand's Education Hub for the Southern Region of the ASEAN Countries Project Office of the Higher Education Commission, as well as by the Natural Rubber Innovation Research Institute (Grant No. SCI6201170s), Prince of Songkla University.

**Conflicts of Interest:** The authors declare no conflict of interest.

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
