# Peer review of "A Self-Healing System Based on Ester Crosslinks for Carbon Black-Filled Rubber Compounds"

_jcs, doi:10.3390/jcs5030070_

Round 1

Reviewer 1 Report

The authors focus on the sacrificial bonds are defined as the weak crosslinks that rapture before the failure of strong covalent bonds. Several material designs based on sacrificial bonds using different bonds and ratios are selected and they are introduced as crosslinking units into rubber based mateirals to synthesize self-healing polymers. Its scientific novelty is well-recognized. That being said, I think that as it stands,  the work contains no apparent serious flaw, several critical issue and minor issues as available all over the manuscript should be addressed before the publication of this manuscript in J. Compos. Sci.  There are many reports out describing in “intro” related to self-healing rubber, but what deficiencies do the authors need to continue to study? From this manuscript, what is the major improvement of your EMZ self-healing modifier?

Author Response

Dear highly learned reviewer,

Thank you for your valuable feedback on our manuscript. Attached pleased find the response to your comments.

Kindly suggest if there are further points to be improved, or if you need more information on a particular issue, feel free to let us know.

Best regards,

Wisut Kaewsakul

University of Twente   

Reviewer 2 Report

The article titled "A self-healing system based on ester crosslinks for carbon black-filled rubber compounds" concerns the preparation of a modifier giving the possibility of the occurrence of self-healing phenomenon in rubber products, which may find application in advanced engineering technologies such as tires. The authors of this publication prepared a suitable modifier, based on a relevant literature review and their previous research, and studied the effect of the modifier on the self-healing phenomenon, dynamic and physico-mechanical properties. The experiment was designed to test each possibility (timing of modifier introduction, filler interaction assessment correlated with self-healing phenomenon, etc.), which shows that the research was planned thoughtfully and does not raise any doubts about its validity. The charts and figures presented are clear, and the description included for them is sufficient. Moreover, the article is written in such a way that the reader easily understands the content and the authors' intentions. The article is of a high standard and does not require significant revision. Congratulations, well done.

Author Response

(The authors gave the same response as above.)

Reviewer 3 Report

Nice attempt of incorporating thermally reversible dynamic networks to filled NR/BR compounds.

The paper is about self-healing system but strangely enough, nothing is mentioned about the temperature dependence of the kinetics of dynamic network formation in EMZ at different temperatures (40-150°C) referred in this paper. Also, it would be interesting to mention the characteristics (tensile, dynamic-mechanical) of pure ENR based self-healing modifier-the EMZ in order to understand effects observed for composites. Please consider.

Mullins effect is a complex phenomenon and relates more to the chain breakage at filler interface, molecular slippage, filler cluster rupture and chain disentanglements, so the self-healing property can more easy be shown by a simple stress-stretch test of unfilled rubbers containing different contents of EMZ. This is in-line with the given topic of self-healing; a more straight forward and widely opted method. Once, the macroscopic self -healing is established, then investigation of molecular or chain level self-healing in filled rubbers would be better understandable and convincing.

The use of two series of compounds i.e. Extra Addition and Blending is not convincingly justified in discussion with large repetition in text.

It is more convincing to show the original curves, may be along with the provided comparative charts.

Author Response

(The authors gave the same response as above.)

Round 2

Reviewer 1 Report

The author carefully addressed and answered all the points. I would recommend publishing this manuscript

Reviewer 3 Report

Further to the author’s reply, please consider that the electronic versions of cited reports [21, 22] are not available to check the characteristics of EMZ modifier, which are important for the presented material system.

Author describes the interchangeability of dynamic ester bonds as the reason of self-healing and shows it by a plausible mechanism in Fig. 7. Considering this, the macroscopic self-healing should be observable in unfilled rubber system: as replied “We did not check the macroscopic level healing, but we expect that we will not observe clear evidence of the healing effect compared to the one without the modifier”.   This reflects that besides the dynamic ester network some additional interactive mechanisms in CB filled NR seems active and responsible for the observed property effects. Please consider.